# Potential Use of Kasedbok (*Neptunia javanica* Miq.) on Feed Intake, Digestibility, Rumen Fermentation, and Microbial Populations in Thai Native Beef Cattle

**DOI:** 10.3390/ani13040733

**Published:** 2023-02-18

**Authors:** Pongsatorn Gunun, Anusorn Cherdthong, Pichad Khejornsart, Sineenart Polyorach, Walailuck Kaewwongsa, Nirawan Gunun

**Affiliations:** 1Department of Animal Science, Faculty of Natural Resources, Rajamangala University of Technology Isan, Sakon Nakhon Campus, Sakon Nakhon 47160, Thailand; 2Tropical Feed Resources Research, Development Center (TROFREC), Department of Animal Science, Faculty of Agriculture, Khon Kaen University, Khon Kaen 40002, Thailand; 3Department of Agriculture and Resource, Faculty of Natural Resources and Agro-Industry, Kasetsart University, Chalermphakiat Sakon Nakhon Campus, Sakon Nakhon 47000, Thailand; 4Department of Animal Production Technology and Fisheries, Faculty of Agricultural Technology, King Mongkut’s Institute of Technology Ladkrabang, Bangkok 10520, Thailand; 5Department of Animal Science, Faculty of Technology, Udon Thani Rajabhat University, Udon Thani 41000, Thailand

**Keywords:** alternative feed, fodder tree legume, protein sources, rumen characteristics, ruminant

## Abstract

**Simple Summary:**

Kasedbok (*Neptunia javanica* Miq.) is a fodder tree legume that grows successfully in a range of tropical and subtropical regions. Kasedbok could be used as a protein source in beef cattle’s concentrate diet, which would lower the cost of feeding. We evaluated the effect of the Kasedbok levels in the diets on feed intake, digestibility, rumen fermentation, and the microbial population in beef cattle. The findings indicated that using up to 240 g/kg Kasedbok in concentrate diets does not affect feed utilization, rumen characteristics, or microbial population, lowering feed costs.

**Abstract:**

This experiment was conducted to determine the influence of Kasedbok (*Neptunia javanica* Miq.) on the feed utilization, rumen fermentation, and microbial population in Thai Native beef cattle. Four animals with a mean body weight of 295 ± 15 kg were randomly arranged in a 4 × 4 Latin square design. There were four treatments, utilizing 0, 80, 160, and 240 g/kg Kasedbok in concentrate. Local feed resources, including cassava chips, rice bran, palm kernel meal, and soybean meal, were utilized to formulate the concentrate diets, which contained between 11.8 and 12.0% crude protein (CP). The trial was conducted for four periods of three weeks each. The first two weeks consisted of an adaptation period, while the final week was a sampling period. The findings of the current study reveal that feed intake, dry matter (DM), organic matter (OM), neutral detergent fiber (NDF), and acid detergent fiber (ADF) digestibility were similar between treatments. In addition, there was no effect of Kasedbok levels on rumen pH, blood urea nitrogen (BUN) concentration, or volatile fatty acid. However, increasing the inclusion level of Kasedbok linearly decreased CP digestibility and ammonia nitrogen (NH_3_-N) concentration (*p* = 0.04). In contrast, the population of fungal zoospores improved significantly (*p* = 0.03), while the bacterial and protozoal counts remained unchanged (*p* > 0.05). Furthermore, when the level of Kasedbok was increased from 0 to 80, 160, and 240 g/kg DM, the cost of concentrate decreased by 4.1, 7.8, and 10.6 USD/100 kg DM, respectively. The results of this experiment suggest that utilizing 240 g/kg of Kasedbok in a concentrated mixture will not affect feed utilization, rumen fermentation, improve microbial population, and reduce feed cost in Thai native beef cattle.

## 1. Introduction

Ruminants that are raised in the tropics depend mostly on seasonal feed sources that lack quality in crude protein (CP) but are high in crude fiber (CF) [1]. High-protein concentrate diets are commonly used as a supplement in livestock feeding to significantly improve animal productivity [2]. Nevertheless, price increases and uncertain availability have induced a search for alternative feeds [3,4]. Therefore, farmers have sought alternative protein sources and utilized local feed resources to reduce feed costs and enhance animal productivity and efficiency [5,6]. Legume trees and fodder shrubs have grown in popularity as supplementary feeds due to their high protein levels, minerals, and plant secondary compounds, containing condensed tannins (CTs) and saponins (SPs) [7,8,9].

Numerous legume trees and shrubs have reportedly been utilized effectively as leaf meal, silage, pellets, and/or fresh cut. Examples of such feeds include the utilization of *Flemingia macrophylla* in dairy cow [10], beef cattle [11], *Cnidoscolus aconitifolius* in dairy bulls [12], *Sesbania sesban* in lamb [8], and dairy cow [13], and *Indigofera tinctoria* L. in beef cattle [6]. Moreover, Kasedbok (*Neptunia javanica* Miq.) is one of the fodder tree legumes that grows successfully in a wide range of tropical and subtropical regions, including Thailand, Indonesia, Malaysia, Cambodia, and Vietnam [14]. Kasedbok is a prostrate, terrestrial shrub with small leaflets and ten-stamen flowers. It inhabits salty mud flats and grassy fields on heavy clay below 650 m in altitude [15]. According to Cruz-Garcia and Price [16], Kasedbok is a kind of wild edible plant that contributes to rice farmers’ food and nutritional security in northeast Thailand. It is widely distributed throughout the region, resulting in a large amount of it along roadsides and in vacant areas, leading to excessive production, especially during the rainy season [17]. According to Gunun et al. [18], Kasedbok harvested at 2, 4, and 6 months contains 25.0%, 20.0%, and 19.0% CP; 56.3%, 64.3%, and 69.4% neutral detergent fiber (NDF); 6.3%, 8.8%, and 12.2% acid detergent fiber (ADF); and 6.2%, 8.4% and 12.2% lignin, respectively. As a result of its high nutritional value, Kasedbok has the potential to substitute soybean meal in diets, lowering feed costs. Anantasook et al. [17] found that Kasedbok meal has the potential to be used as a protein source in concentrate diets and can substitute soybean meal up to 60% in an in vitro study. Nevertheless, there is limited information related to the use of Kasedbok as a feed in an in vivo study. Our hypothesis is that using Kasedbok will enhance rumen fermentation and the microbial population in beef cattle while maintaining feed utilization.

Thus, the purpose of the current trial was to determine the influence of Kasedbok in concentrate on feed utilization, rumen characteristics, and microorganisms in beef cattle.

## 2. Materials and Methods

### 2.1. Animals, Diets, and Experimental Design

The study was conducted at the farm of the Animal Science Department in the Faculty of Technology, Udon Thani Rajabhat University, Udon Thani, Thailand. The trials and plan strictly followed the norms of the Thailand Ethics of Animal Experimentation from the National Research Council (record No. U1-02456-2559). Kasedbok (leaf and stem) was harvested from a Kasedbok pot along a roadside or in a vacant area in Udon Thani province, Thailand. Young stem, leaf, and branch were chopped to 3–5 cm in length. Before adding them to the concentrate, they were sun-dried for three days.

Four Thai native beef cattle with a starting body weight (BW) of 295 ± 15 kg were randomly assigned to one of four dietary treatments in a 4 × 4 Latin square design. Animals were fed diets containing Kasedbok at 0, 80, 160, and 240 g/kg of dry matter (DM) in concentrate. The dietary treatments were formulated according to the WTSR [19] to meet the requirements of beef cattle in Thailand, which contains 12.0% CP and 60.0% total digestible nutrients (TDNs). The animals were fed a concentrate at 0.7% of BW in two equal meals at 07:00 h and 16:00 h, with free water and mineral lick blocks. All animals were given rice straw by allowing 10% of DM to be refused. The list of ingredients and chemical compositions of Kasedbok and concentrate are detailed in Table 1.

This trial was conducted for four periods of three weeks each. The first two weeks consisted of an adaptation period, while the final week was a sampling period. Intake was measured for the first 14 days, and individual feces were collected daily for the last seven days of each period. Mineral blocks and clean water were always accessible.

### 2.2. Feed Cost Analysis

The price of the diet, including Kasedbok, was determined using an input budgeting method according to Serrapica et al. [20]. However, our calculations were based on the standard prices of feed ingredients at the nearby supplier’s gate. The prices of feed were adjusted in accordance with their DM and transformed from THB to USD using 0.0306 THB. The feed prices (USD/kg DM) were as follows: cassava chips at 0.30, rice bran at 0.37, palm kernel meal at 0.34, soybean meal at 1.07, Kasedbok at 0.09, urea at 0.45, molasses at 0.25, mineral and vitamin mixture at 0.61, sulfur at 0.45, and salt at 0.31.

### 2.3. Data Collection and Chemical Analysis

The experimental diets were randomly collected twice a week for DM analysis [21]. During the final week of each period, samples of rice straw, concentrate, Kasedbok, and feces were collected daily. Rectal grab sampling was used to collect a sample of approximately 200 g of feces during the morning feeding (08:00 h). When feces were collected at 3 h intervals, the samples were mixed and utilized as a single sample to estimate nutrient digestibility [22]. The DM, CP, and crude ash were evaluated after drying at 60 °C and milling to 1 mm with a Cyclotech Mill and Tecator [21]. Fiber contents, including those of NDF and ADF, were analyzed according to Van Soest et al. [23]. The CT content of the Kasedbok was evaluated using a modified vanillin-HCl approach based on Burns [24]. SPs were quantified using methanol extraction as modified by Kwon et al. [25] and Poungchompu et al. [26]. 

On the final day of each period, samples of rumen fluid and blood were collected at 0 and 4 h after morning feeding. Approximately 10 mL of blood from the jugular vein was analyzed for blood urea nitrogen (BUN) using the method described by Crocker [27]. Rumen fluid pH was measured immediately, and it was then analyzed for ammonia nitrogen (NH_3_-N) with AOAC [21] and volatile fatty acids (VFAs) according to Mathew et al. [28]. The second portion of the filtered fluid sample was fixed in a sterilized 0.9% saline solution with a 10% formalin solution before being analyzed using total direct counting methods for bacteria, protozoa, and fungal zoospores [29].

### 2.4. Statistical Analysis

All the data were subjected to variance analysis using a 4 × 4 Latin square design utilizing the GLM model technique in the SAS program [30]. The treatment trends were statistically compared using orthogonal polynomial contrasts (linear and quadratic). Tukey’s test was used to detect differences between treatment means, and *p* < 0.05 was considered statistically significant.

## 3. Results

### 3.1. Feed Cost and Chemical Composition of Diets

The cost of feed and the chemical composition of experimental diets are detailed in Table 1. As a result, feed costs ranged above 29.1 to 39.7 USD/100 kg DM. The safe costs for replacing soybean meal and palm kernel meal with 80, 160, and 240 g/kg of Kasedbok were −4.1, −7.8, and −10.6 USD/100 kg DM, respectively. The Kasedbok contains 24.1% of CP, 54.3% of NDF, 32.8% of ADF, 9.2% of CT, and 10.9% of SP. The concentrate was formulated using regional feed resources, which contained between 11.8 and 12.0% CP. The NDF and ADF levels increased following the addition of Kasedbok to the concentrate.

### 3.2. Feed Intake and Digestibility

The results of dietary treatment on feed intake and nutrient digestibilities are presented in Table 2. The intake of rice straw and concentrate, as well as total intake, was similar among treatments (*p* > 0.05). While the increasing levels of Kasedbok decreased CP digestibility linearly (*p* = 0.04), it had no impact on the digestibility for DM, OM, NDF, and ADF (*p* > 0.05).

### 3.3. Rumen Fermentation and Blood Metabolites

Table 3 shows the parameters of ruminal pH, NH_3_-N concentration, and blood metabolites. The results revealed that 4 h after feeding, levels of NH_3_-N concentration decreased (*p* = 0.04). However, Kasedbok levels did not influence rumen pH or BUN concentration (*p* > 0.05).

### 3.4. Volatile Fatty Acid (VFA) Profiles

Table 4 shows the results of the VFA profiles. Rising levels of Kasedbok had no effect on the total amount of VFA, as well as the concentrations of acetic acid, propionic acid, and butyric acid (*p* > 0.05).

### 3.5. Microbial Populations

The results of the microbial populations affecting experimental diets are shown in Table 5. The addition of Kasedbok did not alter the population of bacteria and protozoa, both entodiniomorphs and holotrichs (*p* > 0.05). In contrast, the population of fungal zoospores in cattle-fed Kasedbok concentrate increased linearly (*p* = 0.03) after 4 h of feeding.

## 4. Discussion

### 4.1. Feed Cost and Chemical Composition of Diets

Kasedbok is a tropical tree legume, and its leaves have been consumed as a vegetable [14]. According to Gunun et al. [18], Kasedbok has nutrients with the ability to be utilized as ruminant feed. However, research into the feeding of ruminants has been extremely limited. In the current study, the CP in Kasedbok was 24.1% DM, and the amounts of NDF and ADF were 54.3% and 32.8% DM, respectively. Concentrate diets using Kasedbok as an ingredient show a lower cost, which is the salvation of livestock farmers. Therefore, the main value of using Kasedbok as an alternative protein in the diet could be a reduction in the cost of feeding beef cattle. As a result, we agree with Gunun et al.’s [6] conclusion that including 10% legume tree (*Indigofera tinctoria* L.) in concentrate diets could lower feed costs. 

### 4.2. Feed Intake and Digestibility

Dry matter intake can contribute significantly to livestock productivity and can be affected by a variety of factors, along with chemical composition, and physical and chemical properties [31]. Generally, feed intake and digestibility decrease when fiber content in the diet increases [6,32,33] or supplementation of plants containing high levels (>55 g/kg DM) of CT and SP [34,35,36,37]. This discovery is made clear by the fact that such a point of feed consumption restriction because of gut fill is dependent on fiber content [38,39], or that a decrease in palatability and digestion rate is dependent on CT content in feed [34,40]. In the present study, the nutritive value of the concentrate indicated that the addition of Kasedbok increased the concentration of fiber. However, the increased fiber content in the diet had no effect on rice straw, concentrate, or total feed intake. Moreover, DM intake was not affected by Kasedbok addition, which may be due to the low amount of CT intake at 14.7, 29.4, and 44.2 g/kg DM when inclusion of Kasedbok was at 80, 160, and 240 g/kg DM, respectively. This result was similar to those of Farghaly et al. [8], who discovered that feeding *Sesbania sesban* to growing lambs had no influence on feed consumption, and Totakul et al. [12] found that feeding *Cnidoscolus aconitifolius* leaf pellets to growing crossbred bulls had no impact on feed intake.

In contrast, beef cattle would be less able to digest CP when fed Kasedbok-containing feed. A reduction in CP digestibility was the focus of Gunun et al. [35], who included tannins in goats’ diets. A decrease in CP digestion when Kasedbok is included can be explained by tannin–protein complexes, especially in the presence of alkaline and stable hydrogen bonding with pH values of 3.5–7.0 [41,42]. This action could increase bypass protein by decreasing rumen-digestible protein, causing protein to flow in the small intestine [43]. Therefore, the small intestine is able to absorb high-quality protein from feed [44]. This is consistent with the findings of Barry and Manley [45], who discovered that the CT extract of *Lotus pedunculatus* can increase the post-ruminal transport of nitrogen and amino acids, thus increasing the amount of rumen bypass protein. In addition, the inclusion of CT in the diets may increase the suppression of bloat and daily weight gain in lambs [43].

### 4.3. Rumen Fermentation and Blood Metabolites

This study demonstrated that ruminal pH did not differ between treatments. This pH range (6.6 to 7.0) was suitable for fermentation in the rumen, microbial growth, and the activity of microorganisms [46,47]. However, the NH_3_-N concentration declined linearly when the level of Kasedbok increased. As a result, NH_3_-N concentrations correlate with the digestion of CP in the rumen. This could be due to the CT producing a protein–tannin complex, which decreases the ability of dietary protein digestion and NH_3_-N output throughout the fermentation [48]. Gunun et al. [49] indicated that supplementing *Antidesma thwaitesianum* Muell.Arg. pomace containing CT at 300 g/h/d to dairy cows decreased the amount of NH_3_-N in their rumen. Furthermore, Gunun et al. [35] found that NH_3_-N concentration decreased with increasing the level of plant-containing CT in goats. Nevertheless, the BUN concentration in the current study was normal and comparable to the findings of Bhatta et al. [50] and Phesatcha et al. [51], who supplemented tree leaf pellets with fodder. They discovered that the BUN value was between 10.1 and 15.1 mg/dL. In the present study, BUN ranged from 11.5 to 13.8 mg/dL, indicating that cattle fed a Kasedbok-based diet had no negative effect on BUN.

### 4.4. Volatile Fatty Acid (VFA) Profiles

Total VFAs are the major product of microbial fermentation in the rumen [52], which are evaluated in energy metabolism in ruminants [53]. Many studies have found that the VFA profiles change via various leguminous fodder shrub. For example, Wanapat et al. [54] demonstrated that the amount of propionate was increased by *Crotalaria juncea*, L. silage in beef cattle. Furthermore, Özelçam et al. [55] demonstrated that *Paulownia* spp. leaves increase VFA production. Its concentrations vary depending on many factors, including feed ingredient, nutrients, or plan containing secondary compounds, which can affect feed intake, digestibility, rumen ecology, and passage rate [56,57]. According to Khonkhaeng et al. [47], high-sugar feed can serve as a source of fermentable carbohydrates for the production of propionate. In addition, Gunun et al. [6] discovered that when animals were fed high-fiber legume tree, their propionate levels decreased while their acetate levels increased. Moreover, Gunun et al. [35] suggested that goats fed *Terminalia chebula* Retz. containing CT and SP may also have a rise in propionic acid and a decline in acetic acid. However, in the current study, the total VFA production and VFA profile were not changed by Kasedbok’s addition. These results indicate that the inclusion of 240 g/kg Kasedbok in concentrate is appropriate for rumen fermentation in beef cattle without being affected by fiber content or plant secondary compounds.

### 4.5. Microbial Populations

The increasing level of Kasedbok in the diets had a stimulatory effect on the fungal zoospore population, while the bacteria and protozoa were unchanged. Accordingly, Phesatcha et al. [11] demonstrated that the addition of fodder tree legume (*Flemingia macrophylla*) had a significant impact on the population of fungal zoospores. In contrast, Totakul et al. [12] reported that growing crossbred bulls fed fodder tree (*Cnidoscolus aconitifolius*) leaf pellet up to 8% of DM intake did not change the number of fungal zoospores. The rumen fungi produce high concentrations of cellulases and hemicellulases and are capable of hydrolyzing or solubilizing the entire plant cell wall. In addition, rumen fungi are capable of degrading the more resistant plant walls in roughage [58]. However, although the population of fungal zoospores in this trial increased linearly with the inclusion of Kasedbok, this did not alter fiber digestibility. This could be influenced by various factors, including the proportion of roughage to concentrate, concentration of digestible carbohydrate and protein, and feeding frequency, as reported by Koike and Kobayashi [59] and Kang et al. [60].

## 5. Conclusions

The inclusion of Kasedbok in the concentrate had an effect on the CP digestibility and NH_3_-N concentration. However, increasing the level of Kasedbok did not affect feed intake, pH, BUN, or VFA concentration, while it enhanced fungal zoospore populations and reduced the cost of the concentrate. The addition of 240 g/kg of Kasedbok to the concentrate demonstrated that it could serve as a protein source and maintain cattle production while also lowering the feed cost. However, this study was limited in that it did not examine growth performance, carcass, or meat quality. Further experiments are needed to evaluate the influence of Kasedbok on the growth rate, carcass, and meat quality of beef cattle.

## Figures and Tables

**Table 1 animals-13-00733-t001:** Ingredients and chemical compositions of the diets.

Item	Rice Straw	Kasedbok	Levels of Kasedbok in Diet (g/kg of DM)			
			0	80	160	240
Ingredient, g/kg of DM						
Cassava chip			408	363	361	340
Rice bran			327	321	249	228
Palm kernel meal			111	113	126	98
Soybean meal			85	54	33	22
Kasedbok			0.0	80	160	240
Urea			23	23	23	23
Molasses			27	27	29	30
Minerals and vitamins			9	9	9	9
Pure sulfur			5	5	5	5
Salt			5	5	5	5
Total feeding costs, (USD/100 kg DM)			39.7	35.6	31.9	29.1
Safe costs (vs 0 g/kg Kasedbok), (USD/100 kg DM)			0	−4.1	−7.8	−10.6
Chemical composition						
Dry matter, %	95.2	51.3	95.8	95.8	95.9	95.1
Organic matter, % of DM	87.2	93.4	93.2	93.7	93.6	93.0
Crude protein, % of DM	2.4	24.1	12.0	11.9	11.9	11.8
Neutral detergent fiber, % of DM	75.8	54.3	42.0	50.0	53.0	54.4
Acid detergent fiber, % of DM	41.4	32.8	28.8	31.6	32.6	32.9
Total Digestible Nutrients, % of DM	-	-	60.0	60.0	60.6	60.8
Condensed tannin, % of DM	-	9.2	-	-	-	-
Crude saponin, % of DM	-	10.9	-	-	-	-

**Table 2 animals-13-00733-t002:** Feed Intake and digestibility of Thai native beef cattle.

Item	Levels of Kasedbok in Diet (g/kg of DM)				SEM	Contrast	
	0	80	160	240		Linear	Quadratic
DM intake							
Rice straw							
kg/d	5.5	5.6	5.2	5.2	0.14	0.27	0.89
%BW	1.9	1.9	1.8	1.8	0.04	0.30	0.97
%BW^0.75^	77.3	77.5	73.2	73.1	1.27	0.22	0.95
Concentrate							
kg/d	2.0	2.0	2.0	2.0	0.02	0.53	0.22
%BW	0.7	0.7	0.7	0.6	0.02	0.29	0.14
%BW^0.75^	27.1	27.5	27.5	26.5	0.27	0.42	0.21
Total							
kg/d	7.5	7.6	7.2	7.2	0.11	0.27	0.72
%BW	2.6	2.6	2.5	2.4	0.03	0.20	0.77
%BW^0.75^	104.4	105.0	100.8	100.6	1.38	0.22	0.76
Digestibility coefficients, %							
Dry matter	54.8	54.4	53.3	53.4	0.67	0.45	0.85
Organic matter	58.2	57.8	56.6	56.8	0.76	0.46	0.83
Crude protein	47.1 ^a^	43.0 ^b^	40.4 ^c^	40.8 ^c^	0.55	0.04	0.57
Neutral detergent fiber	53.1	54.6	54.8	54.5	0.36	0.47	0.48
Acid detergent fiber	32.5	31.8	30.0	31.4	0.87	0.41	0.69

^a,b,c^ Values on the same row with different superscripts differed (*p* < 0.05).

**Table 3 animals-13-00733-t003:** Rumen fermentation and blood metabolites of Thai native beef cattle.

Item	Levels of Kasedbok in Diet (g/kg of DM)				SEM	Contrast	
	0	80	160	240		Linear	Quadratic
Ruminal pH							
0 h-post feeding	6.8	7.0	7.0	7.0	0.08	0.31	0.53
4 h-post feeding	6.6	7.0	6.8	6.7	0.11	0.17	0.48
NH_3_-N concentration, mg/dL							
0 h-post feeding	14.3	13.6	12.2	14.9	0.51	0.82	0.14
4 h-post feeding	16.1 ^a^	14.3 ^b^	13.3 ^c^	13.1 ^c^	0.22	0.04	0.20
BUN concentration, mg/dL							
0 h-post feeding	11.8	11.5	13.0	11.5	0.43	0.97	0.50
4 h-post feeding	12.8	12.3	13.8	12.3	0.40	0.87	0.56

^a,b,c^ Values on the same row with different superscripts differed (*p* < 0.05).

**Table 4 animals-13-00733-t004:** Ruminal volatile fatty acid profile of Thai native beef cattle.

Item	Levels of Kasedbok in Diet (g/kg of DM)				SEM	Contrast	
	0	80	160	240		Linear	Quadratic
Total VFA, mmol/L							
0 h-post feeding	120.4	116.9	116.2	118.3	1.41	0.62	0.45
4 h-post feeding	123.7	120.6	118.0	119.1	1.38	0.24	0.12
VFA, mol/100 mol							
Acetic acid							
0 h-post feeding	72.7	73.9	75.0	75.2	0.98	0.31	0.49
4 h-post feeding	70.0	70.2	71.7	72.4	0.78	0.25	0.63
Propionic acid							
0 h-post feeding	17.1	17.4	16.1	16.2	0.46	0.66	0.64
4 h-post feeding	20.2	19.8	18.0	18.0	0.76	0.83	0.47
Butyric acid							
0 h-post feeding	10.2	8.7	8.9	8.6	0.99	0.90	0.42
4 h-post feeding	9.8	10.0	10.3	9.6	0.61	0.76	0.46
Acetic/propionic acid ratio							
0 h-post feeding	4.3	4.2	4.7	4.6	0.21	0.66	0.57
4 h-post feeding	3.5	3.5	4.0	4.0	0.19	0.22	0.76

**Table 5 animals-13-00733-t005:** Microbial population in the rumen of Thai native beef cattle.

Item	Levels of Kasedbok in Diet (g/kg of DM)				SEM	Contrast	
	0	80	160	240		Linear	Quadratic
Direct count, (cell/mL)							
Total bacteria, ×10^8^							
0 h-post feeding	6.9	6.2	6.3	6.1	0.36	0.50	0.77
4 h-post feeding	6.3	6.0	6.5	6.3	0.37	0.93	1.00
Protozoa, ×10^5^							
Entodiniomorph	1.5	1.5	1.8	1.3	0.25	0.18	0.28
0 h post feeding	1.5	1.9	1.7	1.5	0.14	0.23	0.47
4 h-post feeding							
Holotrich							
0 h post feeding	2.5	2.0	2.7	2.0	0.31	0.82	0.19
4 h-post feeding	1.8	1.9	1.5	2.0	0.16	0.54	0.17
Total Protozoa							
0 h post feeding	4.0	3.5	4.5	3.3	0.35	0.45	0.78
4 h-post feeding	3.3	3.8	2.5	3.5	0.47	0.61	0.52
Fungi, ×10^4^							
0 h-post feeding	5.9	6.4	7.7	7.8	0.79	0.39	0.93
4 h-post feeding	6.3 ^b^	6.0 ^b^	7.0 ^b^	8.5 ^a^	0.40	0.03	0.17

^a,b^ Values on the same row with different superscripts differed (*p* < 0.05).

## Data Availability

Not applicable.

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
