# Peer review of "Potential Use of Kasedbok (Neptunia javanica Miq.) on Feed Intake, Digestibility, Rumen Fermentation, and Microbial Populations in Thai Native Beef Cattle"

_animals, 2023, doi:10.3390/ani13040733_

Round 1

Reviewer 1 Report

Potential Use of Kasedbok (Neptunia javanica Miq.) on Feed Intake, Digestibility, Rumen Fermentation and Microbial Populations in Thai-Native Beef Cattle

Dear Authors,

the manuscript is very interesting and describes possibility of application different amount of Kasedbok (Neptunia javanica Miq.) in the concentrate and its effect for feed intake, the digestibility, rumen fermentation and microbial populations in Thai-Native Beef Cattle. In conclusions is emphasized that this component in amount of 24% in  positively affect parameters mentioned above. Useful will be information about the body weight and FCR of cattle, because looking of the crude protein digestibility values for linear contrasts statistical analysis should confirm differences in case of the crude protein apparent digestibility (p-value is equal 0.04). In this case the null hypothesis must be rejected and significant difference could be determined between control and treatments contained Kasedbok in amount of 16 or 24%, difference in crude protein apparent digestibility between control and those treatments is about 7%, what can decrease BW and increase FCR. Especially when the tannins and the saponins consist 20% of used experimental ingredient. In statistical analysis could be also used ANOVA for main effect (that could make easier to add of superscripts in the row of table), but contrast test should be also possible to present differences in case of statistical significant differences. But compensation of growth can be also caused by high-quality protein reach intestine as a consequence of by-pass effect, even in case of 7% difference in apparent digestibility of crude protein (higher biological value of protein, which weren’t utilized by microbials in rumen). Below I add some suggestions helpful during correction of the text (there are not many of them):

Line 50

Part of the sentence: ‘…, and plant secondary compounds containing CT and saponins (SP) [6,7,8]. CT is used first time in the manuscript text, that is why in this case condensed tannins must be added. In this case should be: ‘… and plant secondary compounds containing condensed (CT) tannins and saponins (SP) [6-8].

Line 76

Subsection 2.1. Animals, Diets, and Experimental design.

Line spacing must be change.

Line 86

In the text is: ‘… which contain 12.0% CP and 60.0 total digestible nutrients (TDN).’ should be: ‘… and 60.0 % total digestible nutrients (TDN).

Line 95

Table 1. Ingredient and chemical composition of experimental diets (concentrates).

According to International System of Units percentages could be changed for g‧kg-1.

The chemical composition of rice straw could be also valuable information in Table 1.

Energy level will be also important (metabolic energy, net energy it depends from the system used in requirements for cattle in Thailand).

Line 111

Probably crude ash? according to AOAC 2012.

Line 122

Description of orthogonal polynominal contrast is needed in the tables and comparison of treatments in case of:

·       crude protein apparent digestibility (Table 2, p=0.04, linear function),

·       NH3-N concentration in mg‧dl-1 4 hours post feeding (Table 3, p=0.04, linear function),

·       Funghi count‧ 104 (cell/mL) (Table 5, p=0.03, linear function)

Maybe it is possible to present it like in case one-way ANOVA for main effects using the superscripts and post-hoc test (Duncan, Tukey test) to determine differences between treatments, because in this case difference is confirmed, but there is no description of between which treatments.

Line 143

Information about mean for BW or body weight gains and FCR could be useful. Superscripts are needed in case of crude protein (p=0,04). 0, 8, 16, 24% can be changed for 0, 80, 160 and 240 g‧kg-1.

Line 148

Table 3. The same like in line 143. NH3-N concentration, mg/dl (p=0,04).

Line 161

Table 5. The same like in line 143. Funghi count‧ 104 (cell/mL) (p=0,03) and in case of Holotrich 0h post feeding (Kasedbok in amount of 80 g/kg of concentrate).

Line 197

Line spacing must be changed.

Line 211

In manuscript is mg/dL, it should be mg‧dL-1.

Line 213

Subsection should be in form of italics: 4.4 Volatile Fatty Acid (VFA) Profiles.

Line 247

Line spacing must be changed in paragraph.

Author Response

Dear reviewer,

First of all, the authors highly appreciated all the comments and suggestions made by the reviewers. Above all, the authors felt that all points made were very useful and have incorporated most of the corrections, rewritten sentences and additional data where necessary as suggested in order to make the manuscript ready for possible publication in Animals. All those corrected and modified appeared in yellow color. Please also find all points corrected and modified according to the reviewers’ suggestions.

With the below information we would like to resubmit our paper for your kind considerations for a possible publication in Animals. We again wish to thank you very much for your kind attention and support and looking forward to hearing from you.

Sincerely yours,

Nirawan Gunun

Reviewer 2 Report

Dear authors, interesting manuscript, I have only one major concern, the study has some limitations (n° of animals mainly), I suggest reporting all the limitations in the discussion of the results. Other minor concerns are reported in the attached file.

Author Response

Dear reviewer,

First of all, the authors highly appreciated all the comments and suggestions made by the reviewers. Above all, the authors felt that all points made were very useful and have incorporated most of the corrections, rewritten sentences and additional data where necessary as suggested in order to make the manuscript ready for possible publication in Animals. All those corrected and modified appeared in yellow color. Please also find all points corrected and modified according to the reviewers’ suggestions.

With the below information we would like to resubmit our paper for your kind considerations for a possible publication in Animals. We again wish to thank you very much for your kind attention and support and looking forward to hearing from you.

Sincerely yours,

Nirawan Gunun

Reviewer 2

Dear authors, interesting manuscript, I have only one major concern, the study has some limitations (n° of animals mainly), I suggest reporting all the limitations in the discussion of the results. Other minor concerns are reported in the attached file.

Reply:

Latin square designs are often employed in animal experiments to minimize the number of animals required to detect statistical differences. This design is often employed in studies of ruminant diets, especially those of large ruminants (beef cattle, buffaloes, dairy cows, etc.). At the same period, all animals were tested together with respective treatment and thus, age and health status did not influence animals.  In addition, animals were provided 14 days for adapted to diet for each period, therefore, this standard protocol would allow animal response dietary treatment. As the result, the total data from this design are 16 data and each treatment consisted of 4 replications. Many experiments were use a single 4x4 latin square design (4 animals, 4 treatments and 4 periods) in ruminant published in the high standard international journal such as scientific Reports, Veterinary Sciences, Animal Production Science etc., as well as in Animals. For example:

  1. Dagaew, G., S. Wongtangtintharn, C. Suntara, R. Prachumchai, M. Wanapat and A. Cherdthong. 2022. Feed utilization efficiency and ruminal metabolites in beef cattle fed with cassava pulp fermented yeast waste replacement soybean meal. Scientific Reports. 12:16090. (Impact factor= 4.996; Q1). https://doi.org/10.1038/s41598-022-20471-6

Abstract: The purpose of this study was to see how substituting cassava pulp fermented yeast waste (CSYW) for soybean meal (SBM) in a concentrate affected feed intake, digestibility, and rumen fermentation in Thai native beef cattle. In this study, four male Thai native beef cattle with an average age of 15.0 ± 25.0 months and body weights of 140 ± 5.0 kg were used. The experimental design was a 4 × 4 Latin squared design, with dietary treatments of CSYW replacing SBM at 0, 33, 67, and 100% in the concentrate mixture.

  1. Prachumchai, R., A. Cherdthong, M. Wanapat, S. So and S. Polyorach. 2022. Fresh cassava root replacing cassava chip could enhance milk production of lactating dairy cows fed diets based on high sulfur-containing pellet. Scientific Reports. 12(1):3809 (Impact factor= 4.379; Q1). DOI:10.1038/s41598-022-07825-w

Abstract: The experiment objective was to assess the shifting effect from cassava chip (CC) to fresh cassava root (FC) affected feed utilization, rumen metabolism, cyanide-using bacteria, and milk quality in lactating Thai Friesian dairy cows fed diets based on high sulfur-containing pellet (PS). Four lactating Thai Friesian dairy cows of 481.5 ± 31.3 kg BW (about 4 years old were allocated with four treatments in a 4 × 4 Latin square design. The four treatments were: replacement FC for CC at 0%, 60%, 80%, and 100% dry matter (DM), respectively.

  1. Unnawong, N.; Cherdthong, A.; So, S. Influence of Supplementing Sesbania grandiflora

Pod Meal at Two Dietary Crude Protein Levels on Feed Intake, Fermentation Characteristics, and Methane Mitigation in Thai Purebred Beef Cattle. Vet. Sci. 2021, 8, 35. https://doi.org/10.3390/vetsci8020035

Abstract: The aim of the study was to evaluate the effect of crude protein (CP) levels in concentrate and Sesbania grandiflora pod meal (SG) supplementation on feed intake, rumen fermentation, and methane (CH4) mitigation in Thai purebred beef cattle. Four cattle with 100 ± 5.0 kg body weight were used in this study. A 2 × 2 factorial experiment in a 4 × 4 Latin square design was conducted, in which factor A was the CP levels in concentrate of 14% and 16% of dry matter (DM) and factor B was the supplement levels of SG at 0.4% and 0.6% DM intake, respectively.

  1. Insoongnern, H.; Srakaew, W.; Prapaiwong, T.; Suphrap, N.; Potirahong, S.;Wachirapakorn, C. Effect of Mineral Salt Blocks Containing Sodium Bicarbonate or

Selenium on Ruminal pH, Rumen Fermentation and Milk Production and Composition in Crossbred Dairy Cows. Vet. Sci. 2021, 8, 322. https://doi.org/10.3390/vetsci8120322

Abstract: ………………….. Four crossbred dairy cows with an initial weight of 456 ± 6 kg in mid-lactation were assigned at random using a 4 × 4 Latin square design. The experiments were divided into four periods, each lasting 21 days. Each cow was fed a basal diet supplemented with a different type of mineral salt block: a control with no MSB supplementation, and MSB groups with MSB containing NaHCO3 (MSB-Na), MSB containing Se (MSB-Se), and conventional commercial MSB (MSB-Com). MSB-Na contained NaHCO3 (500 g/kg) to prevent acidosis, MSB-Se contained organic Se (15 mg/kg) as an antioxidant, and MSB-Com was a positive control mineral salt block…………….

  1. Supapong, C.; Cherdthong, A. Effect of Sulfur and Urea Fortification of Fresh Cassava Root in Fermented Total Mixed Ration on the Improvement Milk Quality of Tropical Lactating Cows. Vet. Sci. 2020, 7, 98. https://doi.org/10.3390/vetsci7030098

Abstract: The aim of the present research was to determine the influence of sulfur and urea combined with fresh cassava root in fermented total mixed ration (FTMR) on digestibility, fermentation in the rumen, blood metabolite, milk yield, and milk quality in tropical lactating dairy cows. Four mid-lactation Thai Holstein–Friesian crossbred cows were studied. Pre-experiment milk yield was 12.7 ± 0.30 kg/day, and the body weight was 495 ± 40.0 kg. Animals were evaluated in a 2 × 2 factorial in a 4 × 4 Latin square design to receive diets followed by: factor A, which was a dose of sulfur inclusion at 1.0% and 2.0%, and factor B, which was level of urea inclusion at 1.25% and 2.5% DM in FTMR………..

  1. Maxiselly, Y.; Chiarawipa, R.; Somnuk, K.; Hamchara, P.; Cherdthong, A.; Suntara, C.;

Prachumchai, R.; Chanjula, P. Digestibility, Blood Parameters, Rumen Fermentation, Hematology, and Nitrogen Balance of Goats after Receiving Supplemental Coffee Cherry Pulp as a Source of Phytochemical Nutrients. Vet. Sci. 2022, 9, 532. https://doi.org/ 10.3390/vetsci9100532

Abstract: This research examines the impact of adding dried coffee cherry pulp (CoCP) to goat feed on the digestibility of the feed, rumen fermentation, hematological, and nitrogen balance. A goat feeding experiment employed four male crossbreds (Thai Native x Anglo Nubian) aged 12 months and weighing 21.0 ± 0.2 kg each. The treatment was conceived as a 4 x 4 Latin square with four specific CoCP levels at 0, 100, 200, and 300 g/day……………

  1. Kraiprom, T.; Jantarat, S.; Yaemkong, S.; Cherdthong, A.; Incharoen, T. Feeding Thai Native Sheep Molasses Either Alone or in Combination with Urea-Fermented

Sugarcane Bagasse: The Effects on Nutrient Digestibility, Rumen Fermentation, and Hematological Parameters. Vet. Sci. 2022, 9, 415. https://doi.org/10.3390/vetsci9080415

Abstract: The purpose of this study was to find out how adding molasses to fermented sugarcane bagasse (FSB) alone or in combination with urea affected sheep’s rumen fermentation, hematological parameters, and ability to digest nutrients. Four Thai native sheep with an initial body weight (BW) of 20.87 ± 1.95 kg and 11 ± 1.0 months old were assigned to a 4 x 4 Latin square design with 4 periods of 14-d adaptation and 7 d of sample collection……………….

  1. Darabighane, B.; Tapio, I.; Ventto, L.; Kairenius, P.; Stefa ´ nski, T.; Leskinen, H.; Shingfield, K.J.; Vilkki, J.; Bayat, A.-R. Effects of Starch Level and a Mixture of Sunflower and Fish Oils on Nutrient Intake and Digestibility, Rumen Fermentation, and Ruminal Methane Emissions in Dairy Cows. Animals 2021, 11, 1310. https://doi.org/10.3390
    /ani11051310

Abstract: Four multiparous dairy cows were used in a 4 × 4 Latin square to examine how starch level and oil mixture impact dry matter (DM) intake and digestibility, milk yield and composition, rumen fermentation, ruminal methane (CH4) emissions, and microbial diversity. Experimental treatments comprised high (HS) or low (LS) levels of starch containing 0 or 30 g of a mixture of sunflower and fish oils (2:1 w/w) per kg diet DM (LSO and HSO, respectively)…………………….

  1. Montoya-Flores, M.D.; Molina-Botero, I.C.; Arango, J.; Romano-Muñoz, J.L.; Solorio-Sánchez, F.J.; Aguilar-Pérez, C.F.; Ku-Vera, J.C. Effect of Dried Leaves of Leucaena leucocephala on Rumen Fermentation, Rumen Microbial Population, and Enteric Methane Production in Crossbred Heifers. Animals 2020, 10, 300. https://doi.org/10.3390/ani10020300

Abstract: The effects of dietary inclusion of dried Leucaena leucocephala leaves (DLL) on nutrient digestibility, fermentation parameters, microbial rumen population, and production of enteric methane (CH4) in crossbred heifers were evaluated. Four heifers were used in a 4 × 4 Latin square design consisting of four periods and four levels of inclusion of DLL: 0%, 12%, 24%, and 36% of dry matter (DM) intake……………………

  1. Norrapoke, T.; Tanitpan, P.; Polyorach, S. Cassava pulp can be nutritionally improved by yeast and various crude protein levels fed to cattle. Anim. Prod. Sci. 2022, 62, 333-341. https://doi.org/10.1071/AN20523

Abstract: Dietary supplementation especially feed residues improve by yeast affected rumen fermentation. The aim of the present experiment was to determine the nutritive value, fermentation efficiency and rumen ecology of yeast-fermented cassava pulp, under the use of two levels of protein in concentrate mixtures, in beef cattle. Four beef cattle, 2–3 years of age, were randomly assigned to the following treatments according to a 2 × 2 factorial arrangement in a 4 × 4 Latin square design: cassava pulp fermented either with baker’s yeast or LDD 6 (Factor A), with 16% or 18% CP in concentrate (Factor B).………………………..

  1. The simple summary is very well made
  • Reply: We highly appreciated the compliment from the reviewer.

  1. line 29 please report some other components of the diet.
  • Reply: Already added information, please see in line 30.

  1. line 31 if not significant avoid to report the p value
  • Reply: Already removed, please see in the text.

  1. line 32 also here and below
  • Reply: Already removed, please see in the text.

  1. line 33 so, the protein was more by-pass or undigestable at all?
  • Reply: A reduction in CP digestion when Kasedbok is included is probably due to the tannin-protein complexes it contains. This action could increase bypass protein by decreasing rumen digestible protein, which correlates with the lower concentration of ruminal NH3-N, causing protein to flow in the small intestine.

  1. line 33 please report the precise p value
  • Reply: Already added information, please see in the text.

  1. line 39 i suggest to report keywords not already included in the title
  • Reply: Already improved, please see in the text.

  1. line 43 this statement is too generic
  • Reply: Already improved, please see in the text.

  1. line 46 here i suggest to cite this recent pubblication: 10.3390/ANI12243519

  • Reply: Already added, please see in the text.

  1. line 48 shrubs: what is this?
  • Reply: A fodder shrubs is a woody plant with several perennial stems that may be erect or may stay close to the ground. Numerous fodder shrubs have reportedly been utilized as a alternative protein source for ruminant. Already changed to “fodder shrubs”, please see in the text.

  1. line 50 CT: write extended
  • Reply: Already added, please see in line 55.

  1. line 52-53: very good list
  • Reply: We highly appreciated the compliment from the reviewer.

  1. line 62 i suggest to avoid use the term our but report as impersonal research

  • Reply: Already improved, please see in the text.

  1. line 65 i suggest to analyze also the content of ADL and uNDF
  • Reply: Gunun et al. [17] reported the amount of lignin in Kasedbok, which we already added in the text; however, they did not analyze uNDF.
  1. line 77 please report the ethical approvals
  • Reply: Already added, please see in line 85.

  1. line 87: Rice straw: it is not clear if the feed was chopped or not, if a TMR or separate ingredients, and if you used separate ingredients how did you provided them? timing and technique
  • Reply: Already improved, please see in line 95.

  1. line 93 individually?
  • Reply: Already added, please see in the text.

  1. line 123 Statistical Analysis: did you check for data distribution?
  • Reply: Already correction, please see in line 131.

  1. line 123 i suggest to make other example of research where they used alternative proteins as cost reduction porpose

  • Reply: Already added information, please see in line 178.

  1. line 220. Many factor: also the content of sugars, please add this information an cite: 10.1080/1828051X.2021.1899063.
  • Reply: Already added information, please see in the text.

  1. conclusion: please add the limitations of the study
  • Reply: Already added information, please see in line 263.

  1. Reference: i suggest to report the doi of references
  • Reply: Already added information, please see in the text.

Reviewer 3 Report

This MS deals with the nutritional value of Kasedbok in rations of local beef cattle. It is very well written paper that focus on the use of local rich CP legume trees to reduce the costs and utilize local resourced. However, this report is not enough mature for a full MS and suggest for the authors to submitted as a short communication.

There are some points that needs to be addressed and further be explained.

Author Response

Dear reviewer,

First of all, the authors highly appreciated all the comments and suggestions made by the reviewers. Above all, the authors felt that all points made were very useful and have incorporated most of the corrections, rewritten sentences and additional data where necessary as suggested in order to make the manuscript ready for possible publication in Animals. All those corrected and modified appeared in yellow color. Please also find all points corrected and modified according to the reviewers’ suggestions.

With the below information we would like to resubmit our paper for your kind considerations for a possible publication in Animals. We again wish to thank you very much for your kind attention and support and looking forward to hearing from you.

Sincerely yours,

Nirawan Gunun

Reviewer 3

This MS deals with the nutritional value of Kasedbok in rations of local beef cattle. It is very well written paper that focus on the use of local rich CP legume trees to reduce the costs and utilize local resourced. However, this report is not enough mature for a full MS and suggest for the authors to submitted as a short communication.

There are some points that needs to be addressed and further be explained

Reply: Thank you for your recommendation. However, it has been discovered that many previous studies using 4x4 Latin square designs measured parameters that, similar to this trail (feed intake, digestible, rumen fermentation, and microbial populations in the rumen), could be published as articles. For example:

  1. Dagaew, G., S. Wongtangtintharn, C. Suntara, R. Prachumchai, M. Wanapat and A. Cherdthong. 2022. Feed utilization efficiency and ruminal metabolites in beef cattle fed with cassava pulp fermented yeast waste replacement soybean meal. Scientific Reports. 12:16090. https://doi.org/10.1038/s41598-022-20471-6

Abstract: The purpose of this study was to see how substituting cassava pulp fermented yeast waste (CSYW) for soybean meal (SBM) in a concentrate affected feed intake, digestibility, and rumen fermentation in Thai native beef cattle. In this study, four male Thai native beef cattle with an average age of 15.0 ± 25.0 months and body weights of 140 ± 5.0 kg were used. The experimental design was a 4 × 4 Latin squared design, with dietary treatments of CSYW replacing SBM at 0, 33, 67, and 100% in the concentrate mixture.

  1. Seankamsorn, A., A. Cherdthong, M. Wanapat, C. Supapong, B. Khonkhaeng, S. Uriyapongson, N. Gunun, P. Gunun and P. Chanjula. 2017. Effect of dried rumen digesta pellet levels on feed use, rumen ecology, and blood metabolite in swamp buffalo. Trop. Anim. Health Prod. 49: 79–86. DOI 10.1007/s11250-016-1161-z.

Abstract: The aim of this experiment was to determine the effect of dried rumen digesta pellet levels on feed intake, digestibility, rumen ecology, and blood metabolites in swamp buffalo. Four 2-year-old male swamp buffalo with an initial body weight (BW) of 150 ± 10.0 kg were randomly assigned according to a 4 × 4 Latin square design to receive four levels of dried rumen digesta pellets (DRDPs). The dietary treatments were supplementation of DRDP at 0, 50, 100, and 150 g dry matter/day, respectively. Total feed intake was significantly different among treatments (p < 0.05) and was highest in the 150 g/day DRDP supplement (2.68 kg/day).

  1. Darabighane, B.; Tapio, I.; Ventto, L.; Kairenius, P.; Stefa ´ nski, T.; Leskinen, H.; Shingfield, K.J.; Vilkki, J.; Bayat, A.-R. Effects of Starch Level and a Mixture of Sunflower and Fish Oils on Nutrient Intake and Digestibility, Rumen Fermentation, and Ruminal Methane Emissions in Dairy Cows. Animals 2021, 11, 1310. https://doi.org/10.3390
    /ani11051310

Abstract: Four multiparous dairy cows were used in a 4 × 4 Latin square to examine how starch level and oil mixture impact dry matter (DM) intake and digestibility, milk yield and composition, rumen fermentation, ruminal methane (CH4) emissions, and microbial diversity. Experimental treatments comprised high (HS) or low (LS) levels of starch containing 0 or 30 g of a mixture of sunflower and fish oils (2:1 w/w) per kg diet DM (LSO and HSO, respectively)…………………….

  1. Montoya-Flores, M.D.; Molina-Botero, I.C.; Arango, J.; Romano-Muñoz, J.L.; Solorio-Sánchez, F.J.; Aguilar-Pérez, C.F.; Ku-Vera, J.C. Effect of Dried Leaves of Leucaena leucocephala on Rumen Fermentation, Rumen Microbial Population, and Enteric Methane Production in Crossbred Heifers. Animals 2020, 10, 300. https://doi.org/10.3390/ani10020300

Abstract: The effects of dietary inclusion of dried Leucaena leucocephala leaves (DLL) on nutrient digestibility, fermentation parameters, microbial rumen population, and production of enteric methane (CH4) in crossbred heifers were evaluated. Four heifers were used in a 4 × 4 Latin square design consisting of four periods and four levels of inclusion of DLL: 0%, 12%, 24%, and 36% of dry matter (DM) intake……………………

  1. Norrapoke, T.; Tanitpan, P.; Polyorach, S. Cassava pulp can be nutritionally improved by yeast and various crude protein levels fed to cattle. Anim. Prod. Sci. 2022, 62, 333-341. https://doi.org/10.1071/AN20523

Abstract: Dietary supplementation especially feed residues improve by yeast affected rumen fermentation. The aim of the present experiment was to determine the nutritive value, fermentation efficiency and rumen ecology of yeast-fermented cassava pulp, under the use of two levels of protein in concentrate mixtures, in beef cattle. Four beef cattle, 2–3 years of age, were randomly assigned to the following treatments according to a 2 × 2 factorial arrangement in a 4 × 4 Latin square design: cassava pulp fermented either with baker’s yeast or LDD 6 (Factor A), with 16% or 18% CP in concentrate (Factor B).………………………..

Reviewer 4 Report

Comments and Suggestions for Authors

After reviewing the manuscript entitled “Potential Use of Kasedbok (Neptunia javanica Miq.) on Feed Intake, Digestibility, Rumen Fermentation and Microbial Populations in Thai-Native Beef Cattle”, the following suggestions were made it.

The manuscript, in general, is clear and well organized and provides valuable information on the use of a forage legume that has been little studied. However, some corrections must be made before accepting the manuscript for publication. The main areas for improvement in the manuscript are the use of cubic effects in the statistical analyzes and the need for more clarity in the conclusions section.

Here are some specific comments:

 Abstract

Simple Summary

Line 22: Change “maintains” to “does not affect”.

Abstract

Line 28: Change “liveweight” to “body weight”.

 Line 29: Authors must briefly describe the materials and methods. Here it is important to highlight the number of replicates used per treatment and the duration of the experimental phase.

 Line 30: DM, OM, NDF, and ADF were not previously defined.

 Lines 32 and 33: BUN, CP, and NH3-N were not previously defined.

 Line 35: Please indicate how much the concentrate cost decreased with the inclusion of Kasedbok.

 Line 37: change “in concentrate mixture” to “in a concentrated mixture”.

 Line 37: change “will maintain” to “will not affect”.

 Keywords: Neptunia javanica Miq., rumen fermentation, and Thai-native beef cattle were already used in the title of the manuscript. Please change them for others.

 Introduction

Line 49: change “levels of protein” to “protein levels”.

 Lines 51-52: Please specify on what types of animals these trees and shrubs have been used effectively.

 Line 56: Delete “etc.”

 Lines 59-60: change “the food and nutritional security of rice farmers in northeast Thailand” to “rice farmers' food and nutritional security in northeast Thailand”.

 Lines 60-62: authors should add a reference that supports this information.

 Materials and methods

Line 81: Authors must add information from the ethics committee that approved the procedures used in this study.

 Lines 82: change “weight” to “body weight”.

 Line 81: Authors should add here the information related to the Institutional Review Board or Ethics Committee that approved the animal study protocol.

 Table 1: change “%DM” to “% of DM”.

 Line 108: change “8 a.m.” to “08:00 h”.

 Line 112: The authors must describe the methods used to determine the contents of condensed tannins and crude saponins reported in Table 1.

 Line 125: What was the objective of evaluating the cubic effects? Unfortunately, cubic effects are difficult to explain biologically. Therefore, they must be eliminated from the analysis of all response variables evaluated in this manuscript.

 Line 140: change “groups” to “treatments”.

 Line 145:  change “presents” to “shows”.

 Line 151: change “presents” to “shows”.

 Tables 2-5: The column for cubic effects should be removed.

 Discussion

Line 178-179: The following references may be valuable to deepen the discussion of the effects of condensed tannins on feed intake and digestibility:

 Orzuna-Orzuna, J.F.; Dorantes-Iturbide, G.; Lara-Bueno, A.; Mendoza-Martínez, G.D.; Miranda-Romero, L.A.; Lee-Rangel, H.A. Growth Performance, Meat Quality and Antioxidant Status of Sheep Supplemented with Tannins: A Meta-Analysis. Animals 202111, 3184. https://doi.org/10.3390/ani11113184

 Orzuna-Orzuna, J.F.; Dorantes-Iturbide, G.; Lara-Bueno, A.; Mendoza-Martínez, G.D.; Miranda-Romero, L.A.; Hernández-García, P.A. Effects of Dietary Tannins’ Supplementation on Growth Performance, Rumen Fermentation, and Enteric Methane Emissions in Beef Cattle: A Meta-Analysis. Sustainability 202113, 7410. https://doi.org/10.3390/su13137410

 Line 182-185: What is the relationship between Kasedbok and the plants (Sesbania sesban, Cnidoscolus aconitifolius) used in the studies mentioned here? Its nutritional composition and its content of condensed tannins and saponins were similar?

 Line 232: Delete “in it”.

 Line 236: Why do you compare your results using Kasedbok with those obtained by other authors using Flemingia macrophylla? Are their nutritional profile and secondary metabolite content similar? Please make this clear in the discussion.

 Line 238: Specify which species of fodder trees and at what levels of inclusion in the diet.

 Line 248: Delete “Althought”.

 Conclusions

Lines 249-254: The information in these lines is repetitive. Therefore, the authors should rewrite this information and provide a clearer and more concise conclusion according to the results obtained.

Author Response

Dear reviewer,

First of all, the authors highly appreciated all the comments and suggestions made by the reviewer. Above all, the authors felt that all points made were very useful and have incorporated most of the corrections, rewritten sentences and additional data where necessary as suggested in order to make the manuscript ready for possible publication in Animals. All those corrected and modified appeared in yellow color. Please also find all points corrected and modified according to the reviewers’ suggestions.

With the below information we would like to resubmit our paper for your kind considerations for a possible publication in Animals. We again wish to thank you very much for your kind attention and support and looking forward to hearing from you.

Sincerely yours,

Nirawan Gunun

Reviewer 4

Comments and Suggestions for Authors

After reviewing the manuscript entitled “Potential Use of Kasedbok (Neptunia javanica Miq.) on Feed Intake, Digestibility, Rumen Fermentation and Microbial Populations in Thai-Native Beef Cattle”, the following suggestions were made it.

The manuscript, in general, is clear and well organized and provides valuable information on the use of a forage legume that has been little studied. However, some corrections must be made before accepting the manuscript for publication. The main areas for improvement in the manuscript are the use of cubic effects in the statistical analyzes and the need for more clarity in the conclusions section.

Here are some specific comments:

Reply: We highly appreciated all the comments and suggestions made by the reviewer, and we are carefully making corrections, rewriting sentences, and adding additional data as suggested in order to improve the manuscript. All those corrected and modified appeared in yellow color as below.

  1. Simple Summary

Line 22: Change “maintains” to “does not affect”.

  • Reply: Already changed, please see in line 24.

  1. Abstract

Line 28: Change “liveweight” to “body weight”.

  • Reply: Already changed, please see in line 28.

  1. Line 29: Authors must briefly describe the materials and methods. Here it is important to highlight the number of replicates used per treatment and the duration of the experimental phase.
  • Reply: Already added information, please see in line 31.

  1. Line 30: DM, OM, NDF, and ADF were not previously defined.
  • Reply: Already improved, please see in line 34.

  1. Lines 32 and 33: BUN, CP, and NH3-N were not previously defined.
  • Reply: Already improved, please see in the 32, 36.

  1. Line 35: Please indicate how much the concentrate cost decreased with the inclusion of Kasedbok.
  • Reply: Already improved, please see in line 40.

  1. Line 37: change “in concentrate mixture” to “in a concentrated mixture”.
  • Reply: Already changed, please see in line 42.

  1. Line 37: change “will maintain” to “will not affect”.
  • Reply: Already changed, please see in line 42.

  1. Keywords: Neptunia javanica Miq., rumen fermentation, and Thai-native beef cattle were already used in the title of the manuscript. Please change them for others.
  • Reply: Already improved, please see in the text.

  1. Line 49: change “levels of protein” to “protein levels”.
  • Reply: Already changed, please see in line 45.

  1. Lines 51-52: Please specify on what types of animals these trees and shrubs have been used effectively.
  • Reply: Already added information, please see in line 58.

  1. Line 56: Delete “etc.”
  • Reply: Already delated, please see in the text.

  1. Lines 59-60: change “the food and nutritional security of rice farmers in northeast Thailand” to “rice farmers' food and nutritional security in northeast Thailand”.
  • Reply: Already changed, please see in line
  •  
  1. Lines 60-62: authors should add a reference that supports this information.
  • Reply: Already added information, please see in line 68.

Materials and methods

  1. Line 81: Authors must add information from the ethics committee that approved the procedures used in this study.
  • Reply: Already added information, please see in line 85.

  1. Lines 82: change “weight” to “body weight”.
  • Reply: Already changed, please see in line 89.

  1. Line 81: Authors should add here the information related to the Institutional Review Board or Ethics Committee that approved the animal study protocol.
  • Reply: Already added information, please see in line 85.

  1. Table 1: change “%DM” to “% of DM”.
  • Reply: Already changed, please see in Table 1.

  1. Line 108: change “8 a.m.” to “08:00 h”.
  • Reply: Already changed, please see in line 115.

  1. Line 112: The authors must describe the methods used to determine the contents of condensed tannins and crude saponins reported in Table 1.
  • Reply: Already added information, please see in line 119.

  1. Line 125: What was the objective of evaluating the cubic effects? Unfortunately, cubic effects are difficult to explain biologically. Therefore, they must be eliminated from the analysis of all response variables evaluated in this manuscript.
  • Reply: Already removed, please see in the text.

  1. Line 140: change “groups” to “treatments”.
  • Reply: Already changed, please see in line 149.

  1. Line 145: change “presents” to “shows”.
  • Reply: Already changed, please see in line 154.

  1. Line 151: change “presents” to “shows”.
  • Reply: Already changed, please see in line 159.

  1. Tables 2-5: The column for cubic effects should be removed.
  • Reply: Already removed, please see in the text.

Discussion

  1. Line 178-179: The following references may be valuable to deepen the discussion of the effects of condensed tannins on feed intake and digestibility:

Orzuna-Orzuna, J.F.; Dorantes-Iturbide, G.; Lara-Bueno, A.; Mendoza-Martínez, G.D.; Miranda-Romero, L.A.; Lee-Rangel, H.A. Growth Performance, Meat Quality and Antioxidant Status of Sheep Supplemented with Tannins: A Meta-Analysis. Animals 202111, 3184. https://doi.org/10.3390/ani11113184.

Orzuna-Orzuna, J.F.; Dorantes-Iturbide, G.; Lara-Bueno, A.; Mendoza-Martínez, G.D.; Miranda-Romero, L.A.; Hernández-García, P.A. Effects of Dietary Tannins’ Supplementation on Growth Performance, Rumen Fermentation, and Enteric Methane Emissions in Beef Cattle: A Meta-Analysis. Sustainability 202113, 7410. https://doi.org/10.3390/su13137410 .

  • Reply: Already improved and added information, please see in the text.

  1. Line 182-185: What is the relationship between Kasedbok and the plants (Sesbania sesban, Cnidoscolus aconitifolius) used in the studies mentioned here? Its nutritional composition and its content of condensed tannins and saponins were similar?

Reply: Those of Sesbania sesban and Cnidoscolus aconitifolius are tropical tree legume like Kasedbok which contain low and moderate amount of condensed tannins which may have on feed intake and protein digestibility. Therefore, we refer to those tree legume in the discussion part.

  1. Line 232: Delete “in it”.
  • Reply: Already deleted; please see in the text.

  1. Line 236: Why do you compare your results using Kasedbok with those obtained by other authors using Flemingia macrophylla? Are their nutritional profile and secondary metabolite content similar? Please make this clear in the discussion.
  • Reply: Already added information, please see in line 245.

  1. Line 238: Specify which species of fodder trees and at what levels of inclusion in the diet.
  • Reply: Already added information, please see in line 247.

  1. Line 248: Delete “Althought”.
  • Reply: Already deleted; please see in the text.

Conclusions

  1. Lines 249-254: The information in these lines is repetitive. Therefore, the authors should rewrite this information and provide a clearer and more concise conclusion according to the results obtained.
  • Reply: Already improved; please see in the conclusion.

Round 2

Reviewer 2 Report

Dear authors the paper improved a lot, I have two minor concerns. Please see the attached file

Author Response

Dear reviewer,

First of all, the authors highly appreciated all the comments and suggestions made by the reviewers. Above all, the authors felt that all points made were very useful and have incorporated most of the corrections where necessary as suggested in order to make the manuscript ready for possible publication in Animals. All those corrected and modified appeared in yellow color. Please also find all points corrected and modified according to the reviewers’ suggestions.

With the below information we would like to resubmit our paper for your kind considerations for a possible publication in Animals. We again wish to thank you very much for your kind attention and support and looking forward to hearing from you.

Sincerely yours,

Nirawan Gunun

Reviewer 3 Report

I am sorry this MS is not suitable to be published as a proper scientific paper. 

Author Response

Thank you for your suggestion; however, we have been corrected by three main reviewers who all agreed to publish. Thus, we do hope that you​ will be re-considered our paper to publish.
